# scDSSC: Deep Sparse Subspace Clustering for scRNA-seq Data

**HaiYun Wang[1], JianPing Zhao[1]\*, ChunHou Zheng[2]\*, YanSen Su[2]\***

**1** College of Mathematics and System Sciences, Xinjiang University, Urumqi, China, **2** School of Artificial Intelligence, Anhui University, Hefei, China

\* zhaojianping@126.com (JPZ); zhengch99@126.com (CHZ); suyansen1985@163.com (YSS)

**Data Availability Statement:** All data are in the manuscript and supporting information files. The source code of scDSSC is available at https://github.com/WHY-17/scDSSC.

## Abstract

Single cell RNA sequencing (scRNA-seq) enables researchers to characterize transcriptomic profiles at the single-cell resolution with increasingly high throughput. Clustering is a crucial step in single cell analysis. Clustering analysis of transcriptome profiled by scRNA-seq can reveal the heterogeneity and diversity of cells. However, single cell study still remains great challenges due to its high noise and dimension. Subspace clustering aims at discovering the intrinsic structure of data in unsupervised fashion. In this paper, we propose a deep sparse subspace clustering method scDSSC combining noise reduction and dimensionality reduction for scRNA-seq data, which simultaneously learns feature representation and clustering via explicit modelling of scRNA-seq data generation. Experiments on a variety of scRNA-seq datasets from thousands to tens of thousands of cells have shown that scDSSC can significantly improve clustering performance and facilitate the interpretability of clustering and downstream analysis. Compared to some popular scRNA-deq analysis methods, scDSSC outperformed state-of-the-art methods under various clustering performance metrics.

## Author summary

Single cell RNA sequencing (scRNA-seq) data has been widely used in neuroscience, immunology, oncology and other research fields. Cell type recognition is an important goal of scRNA-seq data analysis, in which clustering analysis is commonly used. However, single cell clustering still remains great challenges due to its high noise, dimension and increasing data scale. Considering the advantages of subspace manifold in processing high-dimensional data and the powerful representation learning ability of deep neural network, we proposed a novel single-cell data clustering method scDSSC, which imitates the generation of scRNA-seq data and reduces the dimension and noise of the data at the same time, and finally outputs the clustering results. Experiments on a variety of scRNA-seq datasets from thousands to tens of thousands of cells have shown that scDSSC can significantly improve downstream analysis, including clustering analysis, cell visualization, differential expression analysis and trajectory inference. In addition, scDSSC has good scalability and can handle large-scale scRNA-seq data.

**Funding:** This work was supported by the open fund of Information Materials and Intelligent Sensing Laboratory of Anhui Province (Grant No. IMIS202105); the Xinjiang Autonomous Region University Research Program (No. XJEDU2019Y002); and the National Natural Science Foundation of China (No. U19A2064, 61873001). The funders had no role in study design, data collection and analysis, decision to publish, or preparation of the manuscript.

**Competing interests:** The authors have declared that no competing interests exist.

## Introduction

Single cell RNA sequencing (scRNA-seq) allows researchers to focus on the transcriptome of a single cell [1,2], which has memorably advanced our knowledge of biological systems by providing unbiased and detailed information to dissect complicated diseases [3]. Clustering analysis groups individual cells into subtypes, such as categorizing subtypes of immune cells for targeted therapy. Due to the sparsity, uncertainty and increasing data scale of single-cell sequencing, scRNA-seq data clustering analysis still faces great challenges.

Most existing clustering methods can be roughly divided into five categories: centroid based, distribution based, connectivity based, graph based, and density based [4], and the above categories are also applicable to the single cell clustering methods. The centroid based clustering methods divide clusters by optimizing a center vector, such as K-means [5]. Some single-cell analysis methods first use PCA [6], tSNE [7], or UMAP [8] algorithms to reduce the dimensionality of the data, and then use classical clustering methods to generate cell clusters, such as pcaReduce [9] (PCA+K-means), SUSCC [10] (UMAP+ K-means) and Seurat (PCA + Louvain) [11]. The distribution based method considers that the data points of each cluster come from a certain statistical distribution, such as Gaussian mixture models (GMM) [12]. The scCNC [13] is a semi-supervised scRNA-seq clustering method, which applied capsule network to reduce dimension, followed by GMM. Density based clustering algorithm examines the connectivity between samples from the perspective of sample density, and continuously expands the cluster based on connectable samples to obtain the final clustering results, such as DBSCAN [14]. Connectivity based methods and density based methods are similar in grouping cell clusters [4]. Graph based methods, such as Louvain [15], maximize the modularity to find communities in the data graph. Louvain is widely used in single cell field, such as scGAE [16], scanpy [17], etc. In addition, Louvain is also integrated into Seurat [11]. Spectral clustering [18] is another algorithm based on graph theory, which is also widely applied in the field of single cell analysis, such as SC3 [19], SIMLR [20], etc.

There is no doubt that a large amount of noise in single-cell RNA sequencing data will significantly affect the results of clustering analysis. The large number of cells profiled via scRNA-seq provides researchers with a unique opportunity to apply deep learning approaches to model the noisy and complex scRNA-seq data. In recent years, many methods based on deep learning have been proposed for noise reduction of scRNA-seq data [21–27]. DCA [21] uses a regular autoencoder to reduce data noise, while scDeepCluster [22] adds Gaussian noise to each encoding layers, and the latter uses deep embedded clustering [28] to generate cell groups. The scDCC [23] is a semi-supervised method, which integrates pairwise constraints into clustering steps. Unlike traditional hard-constrained clustering algorithms, in scDCC, the authors convert (partial) prior knowledge into soft pairwise constraints and add them as additional terms into the loss function for optimization. The pairwise constraint can have two types: must-link (ML) and cannot-link (CL). scDCC, scDeepCluster employ deep embedding clustering to generate final clustering results based on a pre-set number of clusters, which tends to lead to large deviations between the number of cell clusters and the number of real cell types, affecting the clustering performance. To date, there is no method to accurately estimate the number of cell types in scRNA-seq data. Therefore, the number of clusters is set to the true number of cell types in many models as a way to better evaluate the clustering performance, as is the case in this paper.

Due to the existence of noise, it is challenging to identify group structure in high dimensional space [29]. One solution is to perform subspace clustering to achieve optimal multi-subspace representations [30]. Subspace clustering aims to reveal the internal structure of complex data in an unsupervised way [31]. Recently, some new single cell analysis methods

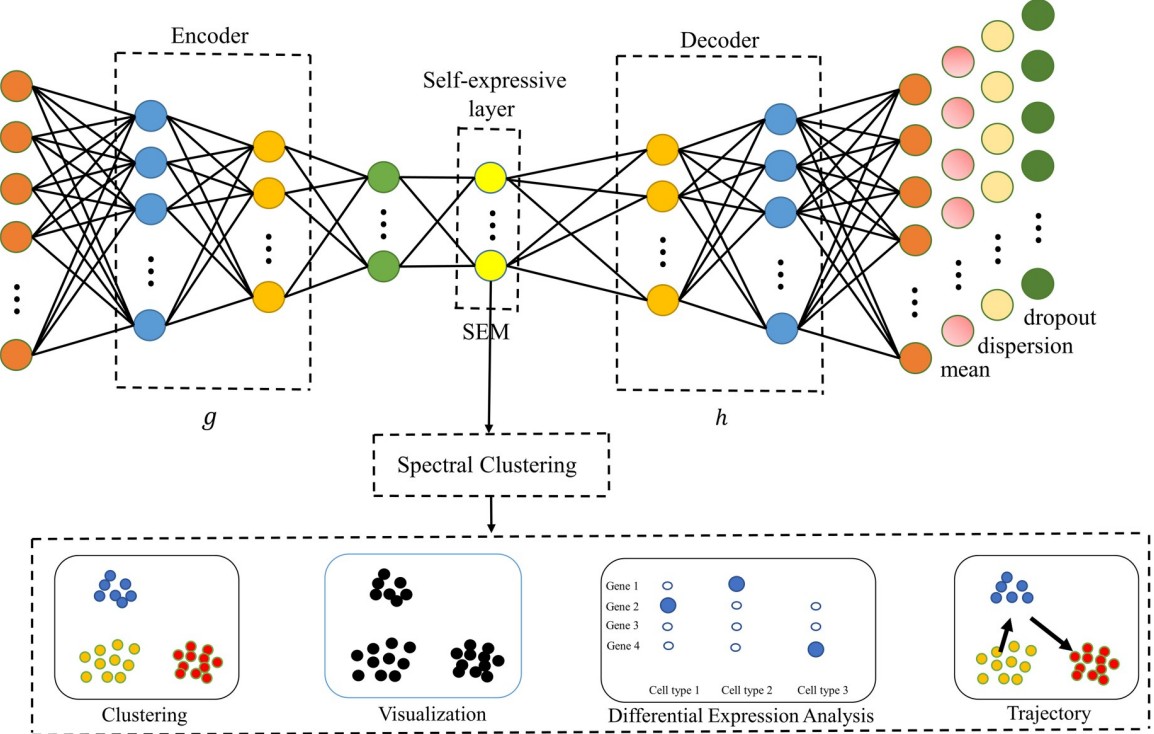

**Fig 1. Network architecture of scDSSC.** An autoencoder based on ZINB distribution is constructed, in which the encoder and decoder can be seen as two nonlinear mappings g and h. The output of the hidden layer is a low dimensional data, which is considered to be the data after dimension reduction and noise reduction. After the last layer of the decoder, four full connection layers are connected, which respectively generate reconstruction data and three parameters in the ZINB distribution. The output of self-expressive layer is self-expressive matrix (SEM), which is sparse and used to perform spectral clustering. According to the clustering results, a series of downstream analysis are applied to verify the performance of the model.

based-subspace clustering have been proposed, including ENCORE [32] and SSRE [33]. ENCORE can distinguish informative features from noise based on feature density profiles, such strategy called as "entropy subspace" separation. However, the computational complexity of ENCORE is too high to be applied to large-scale data. SSRE is based on the hypothesis of subspace to enhance the learning of similarity between cells. SSRE not only has similar limitations to ENCORE, but also does not fully consider the noise and high dimension in scRNA-seq data. In conclusion, most proposed subspace-based models first get the feature representations of scRNA-seq, and then group cells with those representations. However, the feature representation can't completely denote the deep relationships of subspaces, which indicates these models ignore the abundant distribution and manifold information contained in the data.

To overcome the mentioned limitations of existing subspace models, we introduce a deep sparse subspace clustering method scDSSC combining noise and dimensionality reduction for scRNA-seq data, which simultaneously learns feature representation and clustering via explicit modelling of scRNA-seq data generation (Fig 1). Specifically, the main contributions of our study include following parts:

• Different from the subspace clustering method of scRNA-seq data mentioned above, the deep neural network is used for subspace learning of scRNA-seq data, which increases the explicability of deep learning.

- scDSSC fully considers the large amount of noise in scRNA-seq data and the huge number of genes in biological cells, the specific statistical distributions are utilized to simulate scRNA-seq data generation, the autoencoder based on ZINB distribution is applied to reduce the dimension and noise of the data simultaneously.

- Based on the sparsity of scRNA-seq data, we design a sparse subspace clustering model, and get a multi subspace representation of the data. In the mathematical sense, our model has good theoretical support and interpretability.

Experiments on various real scRNA-seq datasets from thousands to tens of thousands of cells have shown that scDSSC outperforms some popular single-cell analysis methods, and significantly improve clustering performance and facilitate the interpretability of clustering and downstream analysis.

## Methods

### Read count data preprocessing and transformation

Firstly, filter out genes that are not expressed in all cells. A gene is expressed in a cell, which means that the gene has a non-zero expression value on the expression profile of that cell.

Secondly, the size factor for every cell is calculated and normalize the read counts by the library size. Before calculating the size factor, the geometric mean of each gene in all cells is first calculated, and the size factor for each cell is the median of the ratio of gene expression to the geometric mean of the gene. The normalized count data can be obtained by following equation:

$$\tilde{x} = \frac{x - \mu}{\sigma}, \tag{1}$$

where $x$ denotes the expression profile of a cell, $\mu$ and $\sigma$ are the mean and standard deviation of all genes in that expression profile, respectively.

Thirdly, take the log transformation and scale the read counts to insure that read counts have unit variance and zero mean.

$$x_{log} = log_2(\tilde{x} + 1). \tag{2}$$

Finally, the first two thousand highly variable genes are selected as the preliminary noise reduction. We apply scanpy [17] to preprocess raw scRNA-seq read count data.

### Zero inflation negative binomial distribution

In this paper, we use Zero Inflation Negative Binomial distribution (ZINB) to model scRNA-seq data generation. Let $X_{ij}$ be the read count for cell $i$ and gene $j$ in scRNA-seq raw count data. ZINB distribution is parameterized by a Negative Binomial (NB) distribution with mean $\mu_{ij}$ and dispersion $\theta_{ij}$, and another parameter $\pi_{ij}$ that represents the probability of dropout of $X_{ij}$.

$$NB\left(X_{ij}; \mu_{ij}, \theta_{ij}\right) = \frac{\Gamma(X_{ij} + \theta_{ij})}{X_{ij}!\Gamma(\theta_{ij})} \left(\frac{\theta_{ij}}{\theta_{ij} + \mu_{ij}}\right)^{\theta_{ij}} \left(\frac{\mu_{ij}}{\theta_{ij} + \mu_{ij}}\right)^{X_{ij}}. \tag{3}$$

$$ZINB(X_{ij}; \pi_{ij}, \mu_{ij}, \theta_{ij}) = \pi_{ij}\delta_0(X_{ij}) + (1 - \pi_{ij})NB(X_{ij}; \mu_{ij}, \theta_{ij}). \tag{4}$$

We will describe how to estimate the three parameters in the ZINB distribution in the

following sections.

## Subspace clustering

Subspace clustering (SC) aims to reveal the internal structure of complex data in an unsupervised way. Currently, representation-based SC (RBSC) method has dominated this field and represents the latest technical level [34]. These methods are based on the hypothesis that a data point can be linearly represented by other data points in the same subspace. This hypothesis is also called Self-Expressiveness Property [34].

**Definition 1 (Self-Expressiveness Property).** Let $X = [x_1, x_2, \cdots x_N] \in R^{d \times N}$ is a collection of $N$ data points, every data point $x_i$ is a $d$-dimensional vector. If each point from a union of independent linear or affine subspaces, each data point $x_i$ can be represented as a linear or affine combination of other points, that is:

$$X = XC, \tag{5}$$

where $C \in R^{N \times N}$ is self-expressiveness matrix (SEM). For scRNA-seq data, the expression profile of a cell can be represented by the expression profiles of other cells predicted to be of the same type. Representing a cell as a linear combination of near-neighboring cells, which tends to capture global structural information, can yield more reliable similarity.

Given the union of multiple subspaces spanned by the cells, the purpose of subspace clustering is to divide them into some category specific groups in an unsupervised way. The self-expression framework aims to calculate SEM and the corresponding affinity matrix, which is then used for spectral clustering. Therefore, the optimization problem of subspace clustering based on self-expression is:

$$min_C \frac{1}{2} \parallel X - XC \parallel_F^2 + f(C),$$
$$s.t. diag(C) = 0. \tag{6}$$

Where $X \in R^{d \times N}$ and $C \in R^{N \times N}$ denote original scRNA-seq read count matrix including d-dimensional N cells and SEM respectively. $f(\cdot)$ is the constraint imposed on C, which can be a combination of one or more norms, e.g., $\ell_0, \ell_1, \ell_2$, rank(C) and so on. Furthermore, sparse subspace clustering can be expressed as:

$$min_C \frac{1}{2} \parallel X - XC \parallel_F^2 + \parallel C \parallel_1,$$
$$s.t. diag(C) = 0. \tag{7}$$

Where $\|C\|_1$ can also be replaced by $\|C\|_2$. Based on the SEM scheme, there have been a variety of works [31–33]. However, it is not necessarily guaranteed that the hand-crafted feature exploit robust descriptions of data samples.

## Deep sparse subspace clustering

Due to the powerful nonlinear feature description of neural network, deep subspace clustering shows good performance. Considering the high dimension of scRNA-seq data, we expect to find a nonlinear mapping to project the original count data into a latent lower dimensional subspace. We introduce a sparse SEM C into neural network framework to achieve deep

 

feature with sparsity preservation. So, solve the optimization problem (5) as follows:

$$min_C \frac{1}{2}||X - \hat{X}||_F^2 + \frac{\lambda}{2}||Y - YC||_F^2,$$
$$s.t. Y = g(X), \hat{X} = h(Y).$$

(8)

Where $Y$ is low-dimensional embedding features, $\hat{X}$ is reconstructed count data, $g$ and $h$ are two nonlinear mapping. In this paper, $g$ and $h$ correspond to Encoder and Decoder in Fig 1 respectively. The first term intents to minimize the reconstruction error and the second term enables the deep embedding features to be updated towards sparsity preservation. In order to ensure the generalization of the model, we add $\ell_2$ constraints to $C$, the optimization problem becomes (9):

$$min_C \frac{1}{2}||X - \hat{X}||_F^2 + \frac{\lambda}{2}||Y - YC||_F^2 + ||C||_2,$$
$$s.t. diag(C) = 0, Y = g(X), \hat{X} = h(Y).$$

(9)

Obviously, up to now, the loss function of the optimization problem consists of the following three parts:

$$L_{reconstruct} = \frac{1}{2}||X - \hat{X}||_F^2,$$

(10)

$$L_{self-expressive} = \frac{\lambda}{2}||Y - YC||_F^2,$$

(11)

$$L_{regression} = ||C||_2,$$

(12)

Although we perform preliminary noise reduction by selecting highly variable genes in the data preprocessing stage, there is still excessive noise in the count data. Therefore, we utilize ZINB distribution mentioned above to model scRNA-seq data generation for further noise reduction.

Let $D = h'_{W'}(g_W(X))$ be the output of the last hidden layer of Decoder, three fully connected layers are employed to estimate the ZINB parameters as follows:

$$M = diag(s_i) \times \exp(W_\mu D),$$
$$\theta = \exp(W_\theta D),$$
$$\pi = sigmoid(W_\pi D).$$

(13)

Where $M$, $\theta$, and $\pi$ represent the matrix form of the estimated mean, dispersion, and dropout probability, respectively. The size factors $s_i$ are precalculated and included as an independent input to the ZINB model-based autoencoder. The loss function of this fraction is the sum of the negative log of ZINB likelihood of all data points.

$$L_{ZINB} = \sum_{ij} -\log(ZINB(X_{ij}; \pi_{ij}, \mu_{ij}, \theta_{ij})).$$

(14)

 

**Table 1. The details of scRNA-seq datasets used in paper.**

| Dataset | Cell number | Gene number | Cell types | Source |
|---|---|---|---|---|
| 10X_PBMC | 4271 | 16653 | 8 | [35] |
| Klein | 2717 | 24175 | 4 | [36] |
| Human_kidney | 5685 | 25215 | 11 | [37] |
| CITE_CMBC | 8617 | 2000 | 15 | [35] |
| Romanov | 2881 | 24341 | 7 | [38] |
| Human1 | 1937 | 20125 | 14 | [39] |
| Human2 | 1724 | 20125 | 14 | [39] |
| Human3 | 3605 | 20125 | 14 | [39] |
| Human4 | 1303 | 20125 | 14 | [39] |
| Mouse1 | 822 | 14878 | 13 | [39] |
| Mouse2 | 1064 | 14878 | 13 | [39] |
| Zeisel | 3005 | 19972 | 9 | [40] |
| HumanLiver | 8444 | 5000 | 11 | [23] |
| Macosko_mouse | 14653 | 11422 | 39 | [41] |

The loss function of scDSSC is:

$$L = (\lambda_1 L_{reconstruct} + \lambda_2 L_{self-expressive} + \lambda_3 L_{regression})^{\frac{1}{10}} + L_{ZINB}$$

$$= \left(\lambda_1 \frac{1}{2}||X - \hat{X}||_F^2 + \lambda_2 \frac{\lambda}{2}||Y - YC||_F^2 + \lambda_3||C||_2\right)^{\frac{1}{10}} + \sum_{ij} -\log(ZINB(X_{ij}; \pi_{ij}, \mu_{ij}, \theta_{ij})).$$

(15)

Where $\lambda_1$, $\lambda_2$ and $\lambda_3$ are weight coefficients used to balance various partial loss functions. In this paper, the three parameters are taken as 0.2, 1.0 and 0.5, respectively. Here, the matrix C is a square matrix whose dimension is equal to the number of cells. In this work, we initialize C using the identity matrix and then continuously update C based on the loss function of the model and the regularized loss imposed on C. Finally, we use the strategy of pre-training and fine-tuning to train the whole model with different learning rates. To demonstrate the necessity of simultaneous $L_{reconstruct}$ and $L_{ZINB}$, several datasets were selected for ablation experiments. The clustering performance under different conditions was evaluated while ensuring that other settings were consistent. The specific numerical results are listed in the S5 Table. The results demonstrate that the model achieves the best clustering performance only when both are in play at the same time.

## The Real scRNA-seq Datasets

The 14 real scRNA-seq datasets are used in this paper, with the number of cells ranging from thousands to tens of thousands. Some datasets are accompanied by obvious batch effect, and there are also large-scale datasets. The details of these datasets are listed in Tables 1 and S1.

## Implementation and parameters setting

The scDSSC model was implemented in Python 3 using PyTorch version 1.6.1. All experiments were conducted on the HP Z840 workstation with 32GB RAM. Here, we introduce the values of some parameters in the network. The encoder consists of three fully connected layers, and the number of neurons is 256-32-10 respectively. The reason why we choose FC layers is that the ordering of genes is arbitrary and fixed in all cells. Unlike natural images, spatially

close pixels exhibit stronger and more complex interdependencies and spatial translation of objects in the image does not usually change their meaning. In the process of training the network, we adopt the strategy of pre-training and fine-tuning, which is completed by setting different learning rates. In this paper, the learning rates of pre-training and fine-tuning are set to 0.002 and 0.001.

### Running the competing methods

Seurat and scanpy are two commonly used scRNA-seq data analysis tools that are widely recognized for their reliable performance and comprehensive functionality. For them, we strictly follow a series of processes given in the official tutorials to get the final results, which include data preprocessing, dimensionality reduction, clustering, etc. When running DCA and scDeepCluster, we followed their original network structure. The low-dimensional embedding of the DCA hidden layer was used as input to K-means, and the number of clusters was set to the true number of cell types to generate clustering results. When using scDCC, the "constraint-pairs" are first computed according to the way proposed in the paper, and then a semi-supervised learning is performed. The specific model can be found in the literature [23]. scGAE performs data preprocessing steps similar to that in this paper, and then performs PCA dimensionality reduction. The reduced-dimensional data is used to calculate the nearest-neighbor graphs of all cells, and the reduced-dimensional computed data and the nearest-neighbor graphs are simultaneously input to a deep network built on the graph autoencoder for model training to generate clustering results. For SSRE, we use the open code, and the parameters in the model also use the optimal parameters recommended in the paper.

## Results

### Overview of the scDSSC Framework

Given an scRNA-seq count data after pre-processing as input, which row represents cell and column means genes, scDSSC imitates the generation of scRNA-seq data and reduces the dimension and noise of the data at the same time, and then learns a self-expressive matrix (SEM). As shown in Fig 1, we design a neural network model similar to the autoencoder, which adds a self-expressive layer behind the hidden layer to learn the SEM. The reconstructed layer is utilized to reconstruct count data. Besides, after the decoder, there are three layers of neural networks, which are used to estimate three parameters in the ZINB distribution. All neural networks are based on the Full Connection (FC) layer.

### Cell segregation via clustering

Here, we demonstrate that scDSSC outperforms state-of-the-art methods in single cell clustering under various clustering performance metrics. We assess the performance of scDSSC in clustering using 14 real scRNA-seq datasets with known cell types that are only used to assess the performance of clustering. Specifically, all real scRNA-seq datasets are derived from specific literature, and these datasets are considered to be well-annotated. The annotated cell types are treated as real labels, which are not used in the computational process of our model and competing models. The output of these models were treated as predicted labels. Finally, we use the information from the true and predicted labels to measure the performance of the model using three external metrics, ACC [42], NMI [43] and ARI [44]. More detailed information about the above three indicators is described in S1 Note. Significantly, we compare scDSSC with 7 scRNA-seq analysis methods widely used in single cell studies: DCA, scDCC, scDeepCluster, scGAE, SSRE, scanpy and Seurat. In addition, SC3 is a good tool for scRNA-

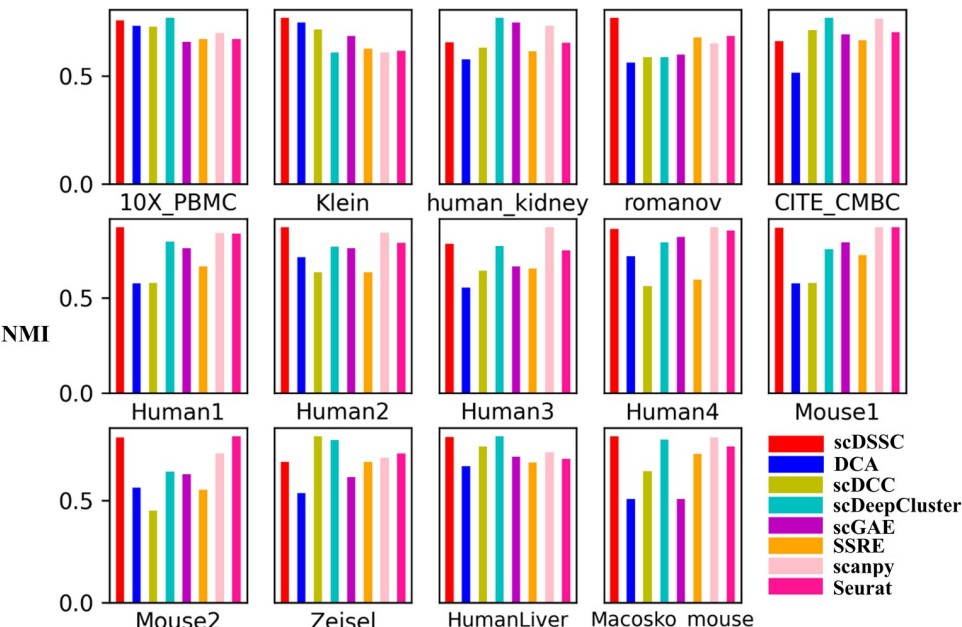

**Fig 2. The clustering performance assessed by NMI.** Each subgraph represents the clustering results of six clustering methods on a dataset. Different colors correspond to different methods, and the ordinate represents the NMI score.

seq data analysis, however it requires a large computational cost. We tested the performance of SC3 with all 14 datasets by borrowing virtual memory, including larger datasets with more than 10,000 cells. We evaluated the clustering results of SC3 on each dataset using both NMI and ARI metrics, and compared SC3 with our method scDSSC, and the specific results are listed in S6 Table. The results in the table show that SC3 outperforms our method scDSSC in clustering on two datasets, Zeisel and Klein, but scDSSC achieves better results on all the remaining 12 datasets.

The scores of the clustering results corresponding to the above seven single cell analysis tools and our method scDSSC on all scRNA-seq datasets under NMI and ARI are listed in S2 and S3 Tables. For example, we have made a more intuitive display of the results under NMI. As shown in Fig 2, scDSSC has the highest clustering accuracy on most datasets, only slightly lower than scDeepCluster on CITE_CMBC and Zeisel datasets. However, the clustering performance of scDSSC on Human_kidney dataset is lower than that of scDeepCluster, scGAE and scanpy, and it is the same under three indicators. The comparison results on ARI and ACC are similar to those on NMI (S1 and S2 Figs). The reason may be that the ZINB distribution does not approximate the true distribution of the Human_kidney dataset well, or the complex structure of this dataset makes it difficult to learn its optimal multi-subspace representation. Briefly, scDSSC outperforms state-of-the-art methods in clustering, and it is a powerful single-cell analysis tool.

## Cell visualization and annotation

Visualizing cells with similar transcripts is helpful to accurately explore the cellular diversity revealed by single-cell transcriptomics and allows researchers to intuitively observe the distribution of cell groups. Only based on the results of cell visualization, we can't directly identify which cell type corresponds to the cell cluster in the biological sense. Therefore, cell annotation is necessary in scRNA-seq data studies.

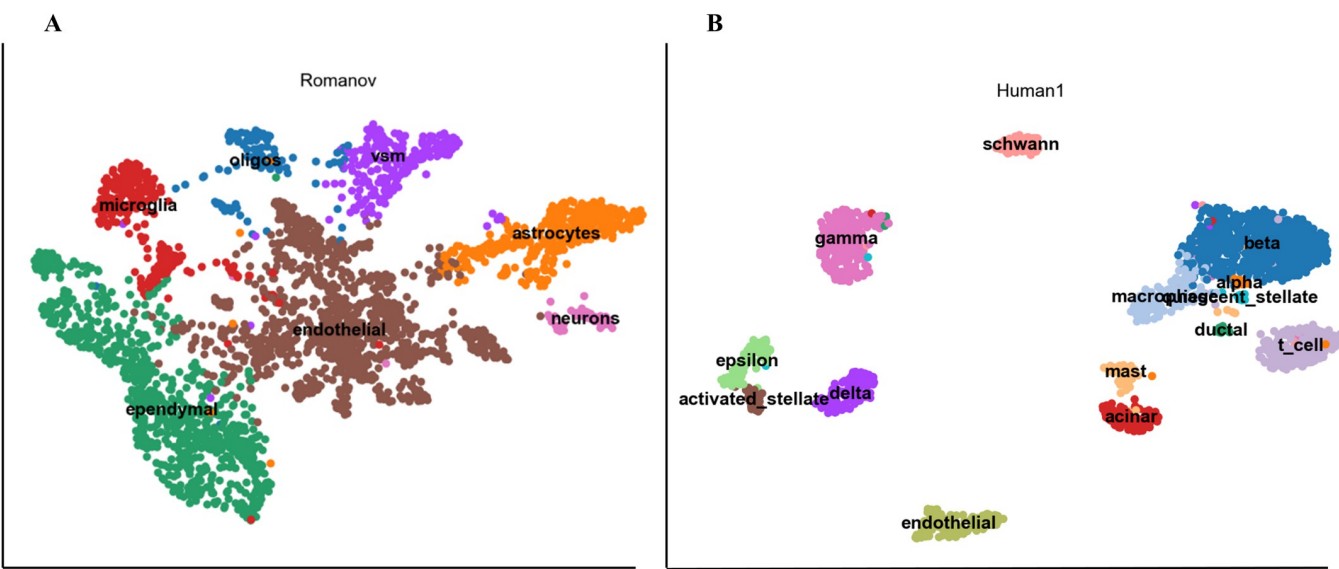

**Fig 3. The results of cell visualization and annotation.** We have given the real cell types on the visualization results, which can help us more intuitively obtain the biological information contained in scRNA-seq datasets. In the figure, we can clearly see the location relationship and distribution between different types of cells. Left A is the visualization result of Romanov dataset, which has 7 cell types. And B represents the results of Human1 dataset, which has 14 cell types.

Here, we select two datasets, Romanov and Human1 with real cell types as the ground-truth labels. Based on the clustering results obtained by scDSSC, scanpy is used for cell visualization and cell annotation. First, the pre-process is performed, the noise and redundant information in the data are removed. Next, calculate the nearest neighbors of each cell according to the given number. This is because before using the UMAP algorithm to reduce dimensions, we must calculate the nearest neighbor graph of each cell, where we must first set a super parameter nearest neighbor number. Finally, UMAP is used to reduce the dimension of data. With low-dimensional embedded data, we can directly apply the existing functions in scanpy to perform cell visualization and cell annotation (Fig 3). For Romanov dataset, there are only microglia cells in which the cells of the same type are slightly dispersed, other types of cells are well aggregated Fig 3A. Turning our attention to the Human1 dataset, scDSSC separated 14 cell types well, where the degree of separation between the different cell types was surprising Fig 3B.

We also used these two datasets as examples and compared the visualization results with the other seven methods. To ensure fairness, the above data preprocessing operation is used for each method of comparison to obtain a two-dimensional representation of the count data. Finally, cell visualization is completed using the clustering results of each method on both datasets. For the Human1 dataset, DCA did not separate the different cell types from each other at all. scDeepCluster, scGAE, Seurat and SSRE showed significant overlap between a few cell types. Comparatively, scDSSC, scDCC and scanpy gave the best results. For the Romanov dataset, the results of scanpy and Seurat are the worst, and the number of cell clusters they delineate far exceeds the number of true cell types in that dataset. The results of the remaining methods on the Romanov dataset showed different degrees of overlapping regions, but relatively less overlapping regions were found for the two subspace learning-based clustering methods, scDSSC and SSRE. More details are shown in S3 and S4 Figs.

### Finding marker genes and cell trajectory inference

In this section, we demonstrate that the accuracy clustering results generated by our method scDSSC can give us support to find marker genes and explore cell trajectory inference. Marker

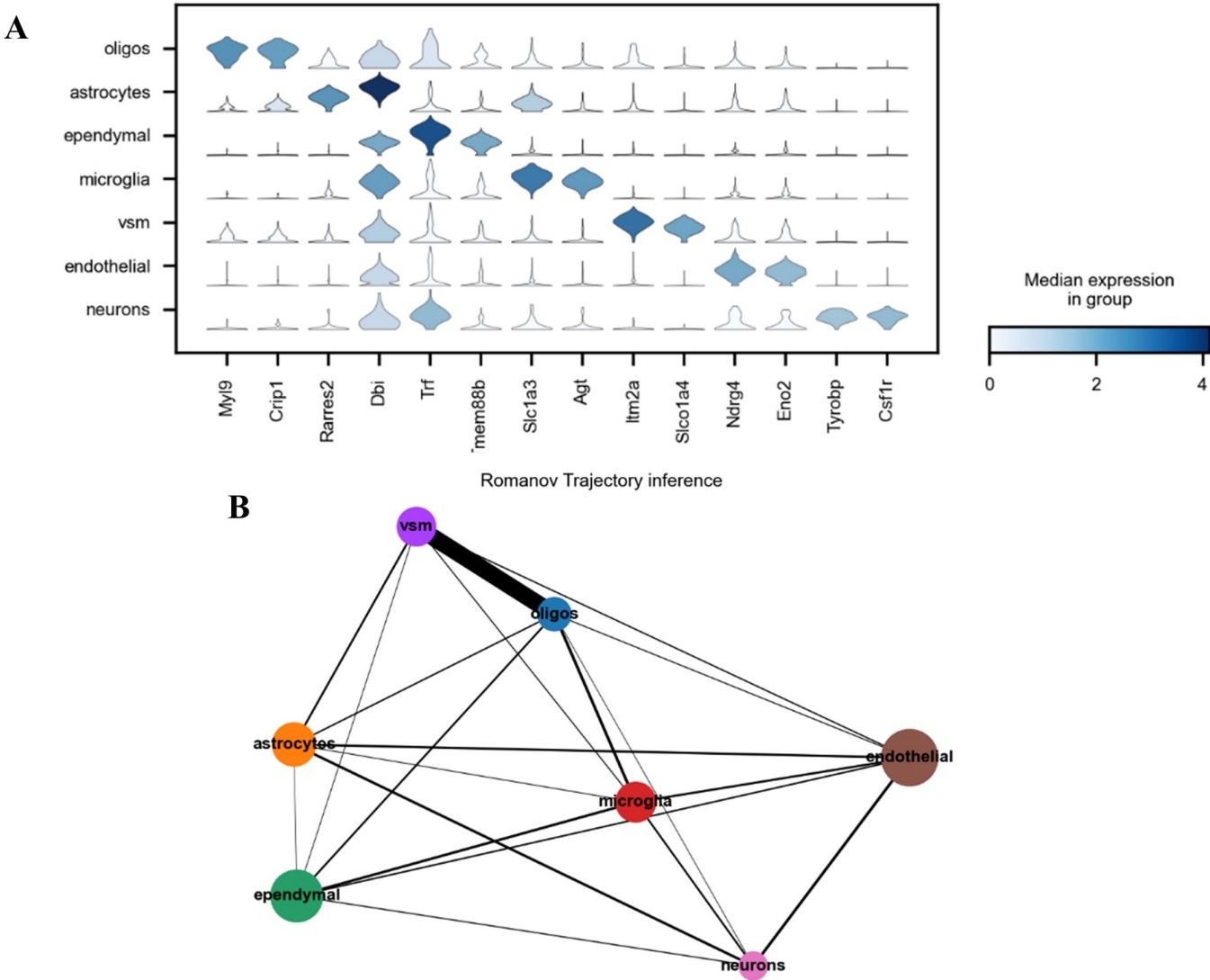

**Fig 4. The results of marker genes and cell trajectory inference.** A shows the distribution of candidate marker gene expression on different types of cells. The left side lists seven cell types, and the lower side lists the marker gene names. The shape of each violin approximates the expression distribution of the marker gene on the cell, and the shade of color reflects the average expression level. B is the result of cell trajectory inference, which reveals the dynamic process of cell differentiation. Seven different colored dots correspond to seven different cell types, and the size of the dots reflects the number of cells.

gene has known function or sequence, which can play the role of specific marker. Now it is widely used in the research of molecular biology, cell biology, developmental biology and so on. The discovery of marker genes is of great significance to explore the mechanism and cure of diseases. Cell trajectory inference also called pseudotime analysis, which order cells along a trajectory based on similarities in their expression patterns. Trajectory inference is a dynamic gene expression model, which can capture the transition state between cell identities and find the continuity of cell differentiation. Here, Romanov dataset is employed as an example to illustrate these (Fig 4).

Use the same data preprocessing steps as in previous section to obtain low-dimensional embedded data. Then we use t-test to calculate the ranking of highly variable genes in each cluster and the first two genes are regarded as marker genes for visualization. Similarly, the trajectory inference results are also obtained from the low-dimensional representation of the original data.

From subgraphs A in Fig 4, we can see the marker genes and their expression corresponding to each cluster. The average expression of marker genes corresponding to each cell cluster is very high, which can be reflected by the color of a single violin in the picture. This means that the marker genes we found have strong differences between different cell types, and the overall results are significant. The subgraph B depicts the dynamic process of cell differentiation, which capture the transition state between cell identities and find the continuity of cell differentiation.

## Scalability and robustness for batch effect

With the rapid development of scRNA-seq technologies, the throughput has increased from initially hundreds of cells to tens of thousands of cells in one run nowadays [45]. Therefore, today's single-cell analysis algorithms are required to handle multiple large-scale scRNA-seq datasets. In addition, an ever-increasing deluge of scRNA-seq data has been generated, often involving different time points, laboratories or sequencing protocols. Batch effect seriously affects downstream analysis, so batch effect correction has been recognized to be indispensable when integrating scRNA-seq data from multiple batches. Fortunately, when the batch effect is smaller than the difference between different cell types, the gene expression profiles of cells within the same subspace can be represented as a linear combination of other cells due to the self-representation property of the subspace, which tends to capture the global structural information and thus curbs the aforementioned random effects on the integrated data.

Here, we demonstrate that scDSSC has good scalability to large-scale datasets and has good robustness to batch effect (Fig 5). According to [39, 41], Human1, Human2, Human3,

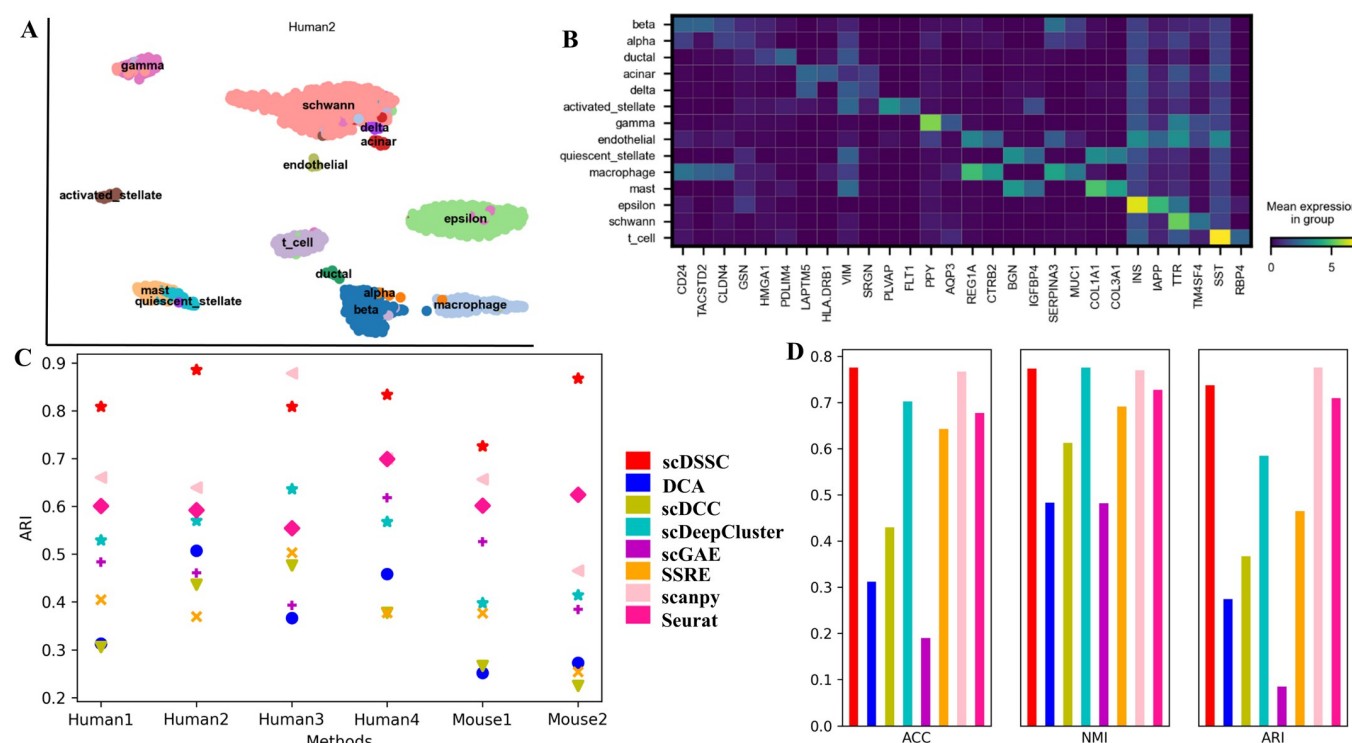

**Fig 5. scDSSC is scalable and robust for batch effect.** Plot A represents the cell visualization result of Human2 dataset, in which the 14 names indicate 14 different cell types. And plot B is the results of differential expression analysis on Human2. The rows and columns indicate the gene names and cell type names, respectively, and the brightness of the color reflects the average expression of the gene. In Plot C, we used ARI to evaluate the clustering performance of eight methods on six small datasets containing batch effects. Plot D shows the performance of scDSSC assessed by ACC, NMI and ARI on a large dataset Macosko_mouse. Plot C and Plot D use the same legend.

Human4, Mouse1, Mouse2 and Macosko_mouse containing batch effect are employed to evaluate the performance of scDSSC in robustness to batch effect. As shown in Fig 5C, in general, scDSSC has the best clustering performance on these datasets containing batch effects, only scanpy performs better than scDSSC on Human3 dataset. For, example, Human2 is used for cell visualization (Fig 5A). The experimental results show that our method makes a clear division of different types. In addition, Human2 has been used for differential expression analysis. Fig 5B basically shows the marker genes for each cluster, except for a few cell types where the expression of the marker genes was not very significant.

In addition, we also demonstrate that scDSSC has good scalability and can handle large-scale datasets. As shown in Fig 5D, scDSSC shows better performance on Macosko_mouse dataset, which consists of 14653 cells. Moreover, we analyze the distribution of cell types and cell differentiation processes in the Macosko_mouse dataset by cell visualization and trajectory inference, respectively (S5 Fig). Due to the limitation of device running memory, we cannot use scDSSC for larger scRNA-seq datasets.

## Simulation datasets

Here, we use several simulated datasets to demonstrate that scDSSC has good performance. Based on the two different dropout rates of 0.05 and 0.25, we generated six separate datasets using Splatter [46], a tool commonly used to generate scRNA-seq simulation datasets. These datasets all consist of two different batches, with six datasets containing five cell types and the remaining six containing two cell types, which also contain unbalanced datasets, that is, there are some types with many cells and others with few cells.

First, we tested the clustering performance of scDSSC on these 12 simulated datasets, and ACC, NMI and ARI were used to evaluate the clustering results (Fig 6A). When the dropout rate was 0.05, scDSSC achieved very good results on all datasets, whether the test dataset contained two cell types or five cell types, or whether it was a balanced dataset or an unbalanced dataset, with all three metrics being 1 on two of the datasets and close to 1 on the remaining dataset. When the dropout rate increased to 0.25, the clustering performance of scDSSC decreased somewhat on an unbalanced dataset containing five cell types. The reason for our analysis is that the number of cells in this unbalanced dataset is small, and the number of certain types of cells is even smaller. Therefore, the sample size is a limitation to learn a multi-subspace representation of that type of cells.

Next, one of the simulated dataset was used to show the results of cell visualization (Fig 6B). What is clear is that we have achieved a precise delineation of the five cell types. In addition, scDSSC effectively overcomes the batch effect contained in the data. We did a differential expression analysis on the same dataset. As can be seen in Fig 6C, we successfully found the corresponding marker genes for each cell type.

## Conclusion and discussion

The rapid development of single cell sequencing technology enables researchers to explore the essence of life activities at the resolution of a single cell. However, high-throughput sequencing also brings great challenges to single-cell data analysis. Due to the existence of noise, it is challenging to identify group structure in high dimensional space. With the rapid development of scRNA-seq technologies, the throughput has increased from initially hundreds of cells to tens of thousands of cells in one run nowadays. Therefore, today's single-cell analysis algorithms are required to handle multiple large-scale scRNA-seq datasets. An ever-increasing deluge of scRNA-seq data has been generated, often involving different time points, laboratories or

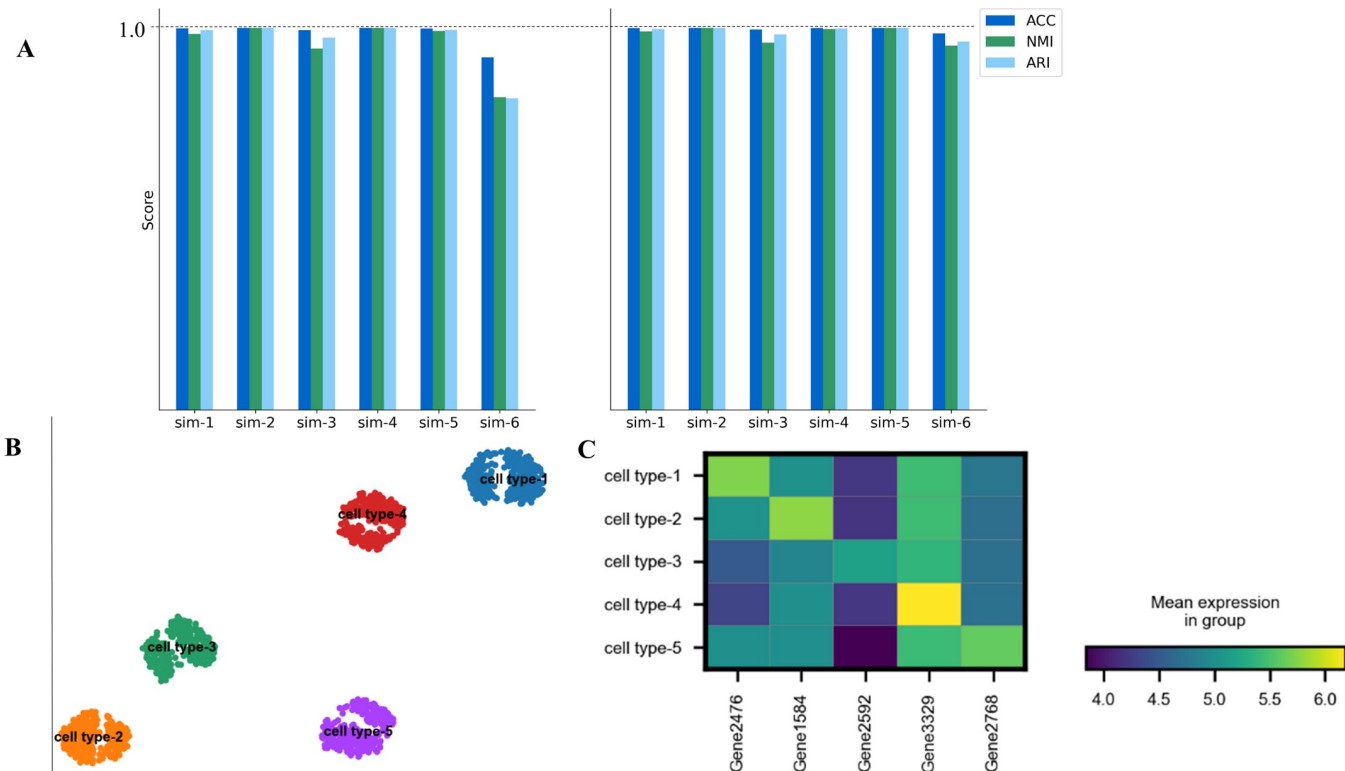

**Fig 6. scDSSC is capable on simulation datasets.** Plot A shows the clustering performance of scDSSC on 12 simulated datasets. The six datasets in the left panel are generated when the dropout rate is set to 0.25, and the six datasets in the right panel are generated when the dropout rate is set to 0.05. Plot B shows the cell visualization results for one of the datasets with five cell types, corresponding to a dropout rate of 0.25. Plot C shows the differential expression analysis for the dataset used in Plot B. The meanings represented by the rows and columns, as well as the meanings represented by the light and dark colors, are consistent with the results of the differential expression analysis described above.

sequencing protocols. Batch effect seriously affects downstream analysis when integrating scRNA-seq data from multiple batches.

In a conclusion, in this paper, we propose a deep sparse subspace clustering method scDSSC combining noise reduction and dimensionality reduction for scRNA-seq data, which simultaneously learns feature representation and clustering via explicit modelling of scRNA-seq data generation. We use 14 real scRNA-seq datasets to test the performance of scDSSC. The results show that scDSSC can generate better clustering results than some other methods, so as to bring satisfying downstream analysis results. In addition, scDSSC can effectively suppress the impact of batch effect on downstream analysis. Significantly, scDSSC can also process large-scale scRNA-seq data generating ideal results simultaneously.

Of course, our model still has some limitations. The training times of each dataset in the pre-training stage and fine-tuning stage are also different. We have listed them in S4 Table. After obtaining the SEM, we need to perform spectral clustering on it to determine the subspace. The dimension of the subspace also needs to be specified, because different subspace dimensions will get different clustering results. Through experiments, we find the optimal subspace dimension corresponding to each dataset, which are listed in S4 Table. There is also the fact that we are not able to apply scDSSC to specific diseases and clinical areas. This is mainly because we have not yet studied specific disease mechanisms, have no researchers in the group who are engaged in clinical research, and have not collected disease-specific single-cell datasets or clinical datasets. In the future work, we will attend to improve these limitations.

## Supporting information

**S1 Note. Evaluation Metrics.** Here we describe ACC, NMI, and ARI in detail, and give their calculation process.
(DOCX)

**S1 Fig. The clustering performance assessed by ARI.** Each subgraph represents the clustering results of eight clustering methods on a dataset. Different colors correspond to different methods, and the ordinate represents the ARI score.
(TIF)

**S2 Fig. The clustering performance assessed by ACC.**
(TIF)

**S3 Fig. The cell visualization results on Human1 and Romanov datasets.** Here, we show the visualization results of scDSSC, DCA, SCANPY and scDCC respectively. The results corresponding to each method are composed of two subgraphs. The left figure represents the visualization results on Human1 dataset, and the right figure represents the visualization results on Romaov dataset.
(TIF)

**S4 Fig. The cell visualization results on Human1 and Romanov datasets.** Here, we show the visualization results of scDeepCluster, scGAE, Seurat and SSRE respectively.
(TIF)

**S5 Fig. The results of cell visualization and trajectory inference on Macosko_mouse dataset.** Plot A shows the cell visualization results, and plot B shows the trajectory inference results.
(TIF)

**S1 Table. The details of datasets used in this paper.** We downloaded the Macosko dataset and filtered cells and genes. Concretely, cells with <700 genes and genes with <3 reads in 3 cells were filtered out. As a result, we obtained 14,653 cells by 11,422 genes among 39 clusters. For CITE_CMBC dataset, we selected the top 2000 dispersed genes to conduct clustering experiments. And for HumanLiver datset, we selected the top 5000 dispersed genes to conduct clustering experiments.
(DOCX)

**S2 Table. The clustering performance accessed by ARI.** The results here correspond to S1 Fig. Here are the ARI scores of eight methods on each dataset.
(DOCX)

**S3 Table. The clustering performance accessed by NMI.** The results here correspond to S2 Fig. Here are the NMI scores of eight methods on each dataset.
(DOCX)

**S4 Table. Parameter setting.** Here, we give the specific parameter values in the process of pre-training and fine-tuning, as well as the value of subspace dimension of different datasets.
(DOCX)

**S5 Table. Loss function ablation experiment.** Three different conditions were set for the whole experiment, namely, the normal scDSSC model, the loss function without MSE and the loss function without ZINB. The metric NMI was used to assess the clustering performance.
(DOCX)

**S6 Table. Comparison of the clustering performance of SC3 and scDSSC on 14 datasets.** (DOCX)

## Author Contributions

**Data curation:** HaiYun Wang.

**Formal analysis:** HaiYun Wang.

**Funding acquisition:** JianPing Zhao, ChunHou Zheng.

**Software:** HaiYun Wang.

**Visualization:** HaiYun Wang.

**Writing – original draft:** HaiYun Wang.

**Writing – review & editing:** JianPing Zhao, ChunHou Zheng, YanSen Su.

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
