## [Decision Letter · Decision Letter 0]

6 Oct 2022

Dear Doctor Zhao,

Thank you very much for submitting your manuscript "scDSSC: Deep Sparse Subspace Clustering for scRNA-seq Data." for consideration at PLOS Computational Biology.

As with all papers reviewed by the journal, your manuscript was reviewed by members of the editorial board and by several independent reviewers. In light of the reviews (below this email), we would like to invite the resubmission of a significantly-revised version that takes into account the reviewers' comments.

As you see from the reports below, the reviewers note that scDSSC performs well but they raise multiple concerns about the manuscript, including regarding references to related work, lack of description of the methods used, and understandability of text and figures.

We cannot make any decision about publication until we have seen the revised manuscript and your response to the reviewers' comments. Your revised manuscript is also likely to be sent to reviewers for further evaluation.

Sincerely,

Maxwell Wing Libbrecht, Ph.D.

Academic Editor

PLOS Computational Biology

Sushmita Roy

Section Editor

PLOS Computational Biology

As you see from the reports below, the reviewers note that scDSSC performs well but they raise multiple concerns about the manuscript, including regarding references to related work, lack of description of the methods used, and understandability of text and figures.

Reviewer's Responses to Questions

**Comments to the Authors:**

Reviewer #1: General

This paper proposes a deep sparse subspace clustering method scDSSC combining noise reduction and dimensionality reduction for scRNA-seq data. Experiments on a variety of scRNA-seq datasets from thousands to tens of thousands of cells have shown that scDSSC can significantly improve clustering performance. My concerns can be found below.

1 You should describe the data preprocessing process in more detail, such as size factor calculation, normalization, log-transformation, etc., and it would be clearer to give the formulae for their calculation

2 In your work, the three hyperparameters λ_1,λ_2 and λ_3 are taken as 0.2, 1.0, 0.5 respectively, is there any special condition for their values? What is the basis for the values you have given them?

3 Is there any connection between pre-training and fine-tuning in the optimization process? Or what are the differences between them?

4 The entire network is built on a fully connected layer, so why not use the more popular convolutional network?

5 In literature, there are already many similar works, e.g. Nucleic Acids Res, 49(D1):D1029-D1037; Bioinformatics. 2021 May 10:btab250, doi: 10.1093/bioinformatics/btab250. The authors should mention related works in their paper.

6 There are some grammatical errors in the manuscript, for example, line 67 “Recently, some new single cell analysis methods based-subspace clustering are proposed”. Please check the entire text carefully and correct any grammatical errors and writing mistakes in the manuscript.

Reviewer #2: In this manuscript, the authors proposed a novel model, scDSSC, for clustering single-cell data. This model employs deep sparse subspace clustering by using a self-expressiveness matrix. The extensive experiments show that scDSSC outperforms most competing methods in multiple real datasets.

Major

L121. The author should explain why a cell can be represented by other cells in a biological view. This is the assumption of this model.

Line 63. “However, the cell clusters obtained by these methods lack the support of mathematical theory”. Need more explanations for this statement. The author should give the mathematical theory which only supports the proposed model.

L168. Could the authors explain why they use both MSE and ZINB as the reconstruction loss. Ablation studies are suggested.

L259. The author should explain why scDSSC can cope with the batch effect in a computational view.

The authors should enhance the resolution of the figures. Many figures are not legible. It is suggested to highlight the differences between scDSSC and scDeepCluster in Figure 1 and 2. From the current figure, I cannot see any differences.

Minor

Line 73. “most proposed models subspace-based” should be ‘most proposed subspace-based model’?

Figure 1 and Figure 2 can be combined

Line 100. The authors should clarify how to calculate size factor and library size.

Line 100-104. Double-check the grammar.

Figure 2. What are the differences between the red and the orange layers? Please specify.

It is suggested to put real data description in the method section.

I suggest the authors to include SC3 in the competing methods

It is suggested to describe the way of running the competing methods in the method section. For example, for using scDCC, how did the authors build the constraints?

Line 211. ‘We infer that the reason may be that the ZINB distribution cannot well approximate the real distribution of Human kidney dataset.’ The authors may try to run the model without ZINB loss.

L 283. This is suggested to put this part into the method section.

I suggest showing NMI in Fig 3 since ACC is influenced by the cluster sizes.

L226. ‘Next, calculate the nearest neighbors of each cell according to the given number’. The author should explain what this step is for.

In the Umap figures, I cannot find any advantages from scDSSC to the competing methods.

L272 ‘’ For, example,’ typo

Reviewer #3: The study by Zhao et al. proposed a deep sparse subspace clustering method scDSSC. Considering the noise and large number of genes in scRNA-seq data, autoencoder based on ZINB distribution is applied to reduce the dimension and noise of the data simultaneously in scDSSC. The authors showed scDSSC outperformed state-of-the-art methods under various clustering performance metrics.

Major concern:

1. Simulated data rather then real data should be used to evaluate the performance of scDSSC, in particular, real data are complexity and could not provide the exactly scenarios under which scDSSC performs better. Furthermore, the clusters in real data are unknown and could not be used to evaluate the performance of these methods. (Fig3–6)

2.scDSSC clustered the cells into subgroups as previous studies (Fig3–6), which did not provide novel biological insight.

3. The authors did not provide information on how to establish the reference for judging the performance of these methods based on real data.

Minor comments:

1.The manuscript is not well organized and should be carefully organized.

2. The fig resolution is low. Some words in fig4 and fig6 could not be recognized.

3. The figure legend is too simple to exactly understand the figure, In particularly Fig4–6.

4. Fig4 should be separated into panel A, B, and C for easy reference.

5. Fig 5A and 5B are redundant.

6. Although all the figures legend are appended at the end of the maintext. It is surprised that the fig legend of Fig 3, 4,5 are also in the maintext, which is redundant.

7. The grammatical errors should be carefully checked.

**Have the authors made all data and (if applicable) computational code underlying the findings in their manuscript fully available?**

Reviewer #1: Yes

Reviewer #2: Yes

Reviewer #3: None

PLOS authors have the option to publish the peer review history of their article (what does this mean?). If published, this will include your full peer review and any attached files.

Reviewer #1: No

Reviewer #2: No

Reviewer #3: **Yes: **Wenfei Jin
---

## [Decision Letter · Decision Letter 1]

14 Nov 2022

Dear Doctor Zhao,

Thank you very much for submitting your manuscript "scDSSC: Deep Sparse Subspace Clustering for scRNA-seq Data." for consideration at PLOS Computational Biology. As with all papers reviewed by the journal, your manuscript was reviewed by members of the editorial board and by several independent reviewers. The reviewers appreciated the attention to an important topic. Based on the reviews, we are likely to accept this manuscript for publication, providing that you modify the manuscript according to the review recommendations.

I am pleased to report that the reviewers found the manuscript to be acceptable for publication, subject to the revisions below. In particular, Reviewer 2 notes several statements that must be clarified.

Sincerely,

Maxwell Wing Libbrecht, Ph.D.

Academic Editor

PLOS Computational Biology

Sushmita Roy

Section Editor

PLOS Computational Biology

I am pleased to report that the reviewers found the manuscript to be acceptable for publication, subject to the revisions below. In particular, Reviewer 2 notes several statements that must be clarified.

Reviewer's Responses to Questions

**Comments to the Authors:**

Reviewer #1: The authors have addressed all my concerns.

Reviewer #2: The authors addressed most of my concerns. Some claims should be further clarified.

1)      The authors state that “the gene expression profile of a cell can be linearly represented by

the gene expression profiles of other cells only within the same subspace. This also means that

these cells belong to the same cell type, which refers to their genotypes being the same and the

expression patterns of genes within the cells are similar.” This statement is misleading because

the cell-type information (ground truth) cannot be used for clustering. Do the authors mean that

a cell is represented by the cells that are predicted to be in the same cell type? Besides, how do

the authors initialize C? This step might be very important for this model.

2)       The authors state that “However, thanks to the self-representation property of subspaces,

the gene expression profile of a cell within the same subspace can be represented as a linear

combination of the expression profiles of other cells, which tends to capture the global structural

information and thus curbs the aforementioned random effects on the integrated data.” I believe

this is only working when the batch effects are smaller than the differences between different

cell types. The authors should specify this in the manuscript.

3)      The authors state that “When using scDCC, the “constraint-pairs” are first computed

according to the way proposed in the paper, and then a semi-supervised learning is performed”.

The performance of scDCC is majorly depending on the constraints used. It is suggested to

specify the way of building constraints, such as the marker gene(s) used as well as the number

of ML and CL links.

4)      If SC3 takes too large RAM or too long times, it is suggested using a subset of cells to run

the experiments.

5)      Some texts on the figures are still too small to read.

Reviewer #3: The manuscript has improved. The authors have addressed my comments in some way.

**Have the authors made all data and (if applicable) computational code underlying the findings in their manuscript fully available?**

Reviewer #1: Yes

Reviewer #2: Yes

Reviewer #3: Yes

PLOS authors have the option to publish the peer review history of their article (what does this mean?). If published, this will include your full peer review and any attached files.

Reviewer #1: No

Reviewer #2: No

Reviewer #3: **Yes: **Wenfei Jin

Figure Files:

Data Requirements:

Reproducibility:

References:

---

## [Editor Report · Decision Letter 2]

28 Nov 2022

Dear Doctor Zhao,

We are pleased to inform you that your manuscript 'scDSSC: Deep Sparse Subspace Clustering for scRNA-seq Data.' has been provisionally accepted for publication in PLOS Computational Biology.

Best regards,

Maxwell Wing Libbrecht, Ph.D.

Academic Editor

PLOS Computational Biology

Sushmita Roy

Section Editor

PLOS Computational Biology

---

## [Editor Report · Acceptance letter]

14 Dec 2022

PCOMPBIOL-D-22-01159R2 

scDSSC: Deep Sparse Subspace Clustering for scRNA-seq Data.

Dear Dr Zhao,

I am pleased to inform you that your manuscript has been formally accepted for publication in PLOS Computational Biology. Your manuscript is now with our production department and you will be notified of the publication date in due course.

With kind regards,

Zsofia Freund
